# Responses of Soil Microbiota to Different Control Methods of the *Spartina alterniflora* in the Yellow River Delta

**DOI:** 10.3390/microorganisms10061122

**Published:** 2022-05-30

**Authors:** Liangyu Li, Xiangyang Jiang, Quanli Zhou, Jun Chen, Yu Zang, Zaiwang Zhang, Chen Gao, Xuexi Tang, Shuai Shang

**Affiliations:** 1College of Marine Life Sciences, Ocean University of China, Qingdao 266005, China; liliangyu98@126.com (L.L.); chenjun271@163.com (J.C.); yuzang@outlook.com (Y.Z.); 2Shandong Provincial Key Laboratory of Marine Ecological Restoration, Shandong Marine Resource and Environment Research Institute, Yantai 250299, China; jxy0535@126.com (X.J.); jiaheyuan15@163.com (Q.Z.); chen_gv@163.com (C.G.); 3College of Biological and Environmental Engineering, Binzhou University, Binzhou 256601, China; zzwangbzu@163.com

**Keywords:** invasion, control method, ITS2 rDNA, marine ecosystems, 16S rRNA

## Abstract

*Spartina alterniflora* invasion has negative effects on the structure and functioning of coastal wetland ecosystems. Therefore, many methods for controlling *S. alterniflora* invasion have been developed. *S. alterniflora* control methods can affect plant community, which results in changes in microbial communities and subsequent changes in soil ecological processes. However, the effects of controlling *S. alterniflora* on soil microbial communities remain poorly understood. We aimed to examine the responses of bacterial and fungal communities to invasion control methods (cutting plus tilling treatment: CT; mechanical rolling treatment: MR). Soil bacterial and fungal community diversity and composition structure were assessed using high-throughput sequencing technology. The findings of the study showed that bacterial diversity and richness in the CT treatment reduced substantially, but fungal diversity and richness did not show any remarkable change. Bacterial and fungal diversity and richness in the MR treatment were not affected considerably. In addition, the two control methods significantly changed the soil microbial community structure. The relative abundance of bacteria negatively associated with nutrient cycling increased considerably in the CT treatment. The considerable increases in the relative abundance of certain bacterial taxa in the MR treatment may promote soil nutrient cycling. Compared with mechanical rolling, soil bacterial community diversity and structure were more sensitive to cutting plus tilling.

## 1. Introduction

Coastal wetlands, transition zones between marine ecosystems and terrestrial ecosystems, provide important ecosystem services function and economic values. Coastal wetland ecosystems were previously reported to be vulnerable to *Spartina alterniflora* invasion, which poses a threat to the ecological balance of these systems [1,2]. *S. alterniflora*, which is native to the gulf coast of the United States, is believed to be the most effective invasive species in the genus *Spartina*, can adapt to harsh coastal environments, and has been introduced around the world [3]. In addition to having a developed root system, this species exhibits rapid growth rates and high biomass production. In view of this, *S. alterniflora* was introduced in China in 1979 to stabilize coastal mud flats and thereby protect China’s coastal wetlands [4,5]. However, the successful introduction of *S. alterniflora* had unexpected and detrimental impacts on the structure and functioning of coastal wetland ecosystems. Owing to its high reproductive capacity, *S. alterniflora* expanded rapidly, replacing native plants, and quickly became a biocontamination in coastal areas, leading to a reduction in the biodiversity of coastal ecosystems [6]. Previous studies have demonstrated that *S. alterniflora* invasion affected soil carbon and nitrogen sequestration and turnover, greenhouse gas emissions, and soil physicochemical characteristics in wetlands, likely because *S. alterniflora* produces more biomass and possesses higher net primary productivity than its native competitors [7,8]. Moreover, *S. alterniflora* invasion can impact the composition and structure of coastal soil microbial communities [9,10]. The Yellow River delta is the most well-preserved and youngest wetland ecosystem in the warm temperate zone in China and occupies an important strategic position in China’s ecological security. Its abundant biological resources provide a key habitat and breeding environment for benthic animals and birds, which thus supports high biodiversity [11,12]. However, the Yellow River delta has been plagued by *S. alterniflora* invasion for several years, mainly manifested by its vigorous colonization of native species habitat on a large scale, habitat fragmentation, and decreased growth and health of native species [13,14]. For protecting the biodiversity and maintaining the ecosystem functioning of the Yellow River delta, effective *S. alterniflora* control measures are required.

Soil microorganisms are an important component of coastal wetland ecosystems and constitute one of the most complex and diverse biological communities on earth. Although soil microorganisms comprise a variety of groups of microscopic life forms, which also include major taxa, such as protists, archaea, and viruses, bacteria and fungi are the most well-studied members [15]. Of note, microorganisms in soil are closely correlated with nutrient cycling, organic matter decomposition, soil functional stability, and plant growth, all of which play a critical role in maintaining the health of coastal wetland ecosystems [16,17,18]. An increasing number of studies have demonstrated that soil microbial communities can be affected by a broad range of biotic and abiotic factors [9,16,19]. For instance, above-ground plants may exert effects on soil microbial communities via direct pathways. Plant species differ in the quality and quantity of their litter and root exudates, and thus promote different soil microbial community compositions and abundance [20,21]. Furthermore, because microorganisms are highly sensitive to soil environmental factors, microbial growth and richness are affected by a fluctuation in these environmental factors [22]. For instance, pH is considered one of the essential soil properties driving bacterial communities. Fierer and Jackson demonstrated that neutral soil was suitable for the growth of bacteria and that acidic soil lowered their diversity [23]. Wu et al. demonstrated that increased pH could promote the growth of gram-negative bacteria and reduce the biomass of gram-positive bacteria [24]. Soil moisture can affect the availability of carbon and nitrogen by regulating liquid diffusion rates, which further affects the microbial community structure [25]. Thus, soil microorganisms play an important role in the ecological processes of wetland soils and can be utilized as biological indicators of soil health.

At present, the most common *S. alterniflora* control methods used globally have been primarily proposed from physical, chemical, and biological perspectives [26,27]. Compared with biological and chemical control methods, physical control methods generally involve the application of mechanical devices to remove or kill *S. alterniflora*; thus, these methods do not pollute or introduce potentially new invasive species into the local environment. However, because a single physical control method is generally inefficient, the integration of two or three different physical methods is preferred [26]. Although integrated physical control methods have demonstrated effective eradication of *S. alterniflora* or inhibition of its germination in subsequent years, their effects on the soil ecosystem and its functioning are not clear [28,29]. Invasion control methods can affect the above-ground plant community, which could modify the soil microbial community composition and structure and subsequently affect general soil processes. Understanding the changes in soil microbial communities following the application of invasion control methods is therefore essential; however, studies examining this aspect have been few. Thus, we selected the Yellow River delta as the study site and used high-throughput sequencing technology to study the effects of two *S. alterniflora* control measures (cutting plus tilling and mechanical rolling) on soil microbial community composition and diversity. The results of this study may provide a microbiological basis for evaluating the impact of control measures on soil ecosystems.

## 2. Materials and Methods

### 2.1. Study Site and Sampling

The study area is located in the Yellow River delta (37°16′ N–38°16′ N, 118°20′ E–119°20′ E). It has a temperate semi-humid continental monsoon climate, which is characterized by four distinct seasons. The mean annual temperature is 11.5 °C–12.4 °C and the mean annual rainfall ranges from 530 to 630 mm with 70% rainfall observed in summer [30]. The soil type is mainly saline soil, and this area is sequentially dominated by *Suaeda salsa*, *Phragmites australis* (Cav.) Tran. ex Stead. and *Tamarix chinensis* Lour. from seaside to land [16].

The experimental site was set at the estuary of Guangli River in Dongying, Shandong Province (Appendix A). *S. alterniflora* density was 80–110 per square meter. Three treatment blocks were set up to compare and analyze the effects of different *S. alterniflora* control methods on soil microbial communities. The three blocks included CT (cutting plus tilling treatment), MR (mechanical rolling treatment), and SC (statistical control without any real treatment). The treatments, including cutting plus tilling and mechanical rolling, were first carried out in early February 2021, and were strengthened in May. In the cutting plus tilling treatment, the above-ground plants of *S. alterniflora* were cut, and then below-ground roots were turned over using ploughing boats. In the mechanical rolling treatment, soils were disturbed by a lightweight tracked vehicle and then above-ground plants of *S. alterniflora* were dislodged and buried in the soil. Samples were collected one week after the end of the treatment (the condition of the treated sites stabilized). There were 10 plots built in the CT, SC, and MR treatment blocks, respectively. The plot was 1 m × 1 m and the distance between adjacent plots in the same treatment block was 100 m. Visible plant litter and stones were removed, and then soil samples at 0–10 cm layer were collected from each plot using the soil auger. A total of 30 samples were obtained. After collection, all samples were immediately placed in a box containing sufficient dry ice and brought back to the laboratory. Samples were stored at −80 °C for subsequent DNA extraction and microbial analysis.

### 2.2. DNA Extraction and Sequencing

Total genomic DNA was extracted from the soil samples using the E.Z.N.A.^®^Soil DNA Kit (D4015, Omega, Inc., Norwalk, CT, USA) as per the manufacture’s instruction. The extracted DNA purity and quality were assessed using an ultraviolet spectrophotometer and agarose gel electrophoresis. The V3–V4 region of the 16S ribosomal RNA gene (for bacteria) was amplified using the primers 341F (5′-CCTACGGGNGGCWGCAG-3′) and 805R (5′-GACTACHVGGGTATCTAATCC-3′), and the ITS2 region of the fungal rRNA gene was amplified using the primers ITS1FI2 (5′-GTGARTCATCGAATCTTTG-3′) and ITS2 (5′-TCCTCCGCTTATTGATATGC-3′) [31]. Polymerase chain reaction (PCR) was performed using the following parameters: an initial denaturation at 98 °C for 30 s, 32 cycles of denaturation at 98 °C for 10 s, annealing at 54 °C for 30 s, extension at 72 °C for 45 s, and final extension at 72 °C for 10 min [31]. The PCR reaction mixture (25 µL) consisted of 12.5 µL of Phusion Hot 2 × Master Mix, 2.5 µL of each primer, and 50 ng of template DNA. There were three replicates for each sample. After amplification, the PCR products of each sample were mixed, and the presence of PCR products was verified with 2% agarose gel electrophoresis. Next, the PCR product was purified using AMPure XT Beads (Beckman Coulter Genomics, Danvers, MA, USA), quantified using Qubit (Invitrogen, Waltham, MA, USA), and then submitted for library preparation. Next, 2 × 250 bp paired-end sequencing was performed on the Illumina NovaSeq platform. Amplicon sequencing and library construction were performed by Lc-Bio Technologies Co., Ltd. (Hangzhou, China). Raw sequence data in the present study were deposited at the Sequence Read Archive (SRA) database of NCBI under accession number (SAMN24054106-24054130 and SAMN24113487-SAMN24113500) and accession number (PRJNA789127 and PRJNA789435).

### 2.3. Sequence Data Processing and Statistical Analysis

Paired-end reads were assigned to samples based on their unique barcodes, and barcode and primer sequences were removed and trimmed. Raw reads were merged using the FLASH software [32]. High-quality clean tags were obtained after denoising and filtering out low-quality reads and chimeric sequences from raw reads using the function “fqtrim”, retaining sequences >1000 bp in length. Singletons’ amplicon sequence variants (ASVs) were removed and an ASV table was generated using DADA2 [33]. Bacterial and fungal sequences were classified using the SILVA v132 database [34]. Alpha and beta diversity of the samples were calculated using QIIME2 [35]. Alpha diversity, represented by Shannon and Chao1 diversity indices [36,37], was used to analyze the complexity of species diversity. Beta diversity was evaluated to analyze the similarity in microbial community structure across all samples. All figures were constructed using R v3.4.4 [38,39]. Differences in microbial alpha diversity and the relative abundance of dominant phyla and genera were tested using the Kruskal–Wallis test and differences in microbial beta diversity were tested using permutational multivariate analysis of variance [40]. Venn diagrams were constructed to visualize the shared and unique ASVs among the three treatments. Principal coordinate analysis (PCoA) with unweighted UniFrac distance was used to visualize the difference in the microbial community structure and composition, respectively, between the three treatments. LDA effect size (LEfSe) was used to show the significant difference of bacterial and fungal communities at different taxonomic level between different treatments. The criterion for LEfSe was set as *p* < 0.05, LDA > 3.0.

## 3. Results

### 3.1. Bacterial Diversity and Composition

The bacterial diversity did not differ significantly between the MR and the SC treatments, but there were significant differences between them and the CT treatment. In the present study, alpha diversity was estimated using Chao1 and Shannon indices. Among the three treatments (Figure 1), Shannon and Chao1 indices were significantly lower (*p* < 0.05) in the CT treatment than in the other two treatments. However, no significant difference was noted between the SC treatment and the MR treatment in terms of diversity and richness.

For bacteria, the number of shared ASVs among the three treatments was 2285, and the number of unique ASVs in CT, SC, and MR treatments were 4223, 10,797, and 13,341, respectively (Figure 2). Compared with the SC treatment, the number of unique ASVs in the CT treatment decreased and that in the MR treatment increased.

PCoA using unweighted UniFrac distance was performed to identify the variation in microbial community composition. PCoA1 explained 10.45% of the variation and PCoA2 explained 8.71% of the variation (Figure 3). In total, 19.16% of variation was explained by the two principal coordinates. The results of PCoA showed that soil bacterial community structure among the three treatments was significantly different (PERMANOVA test, *p* < 0.05), demonstrating that cutting plus tiling and mechanical rolling could shape the bacterial community.

To explore differences in bacterial community composition under different *S. alterniflora* control methods, we compared the relative abundance of bacterial communities for the three treatments. The stacked bar chart (Figure 4) shows the top 10 species in terms of relative abundance. The dominant bacterial phyla in the CT, SC, and MR treatment were Proteobacteria (50.03–57.78%), Bacteroidetes (6.17–16.57%), Chloroflexi (6.13–7.25%), Acidobacteria (3.70–6.98%), Epsilonbacteraeota (2.68–6.58%), Actinobacteria (2.35–3.10%), Planctomycetes (1.56–1.90%), Gemmatimonadetes (1.36–2.92%), and Verrucomicrobia (1.17–2.60%). Proteobacteria was the most predominant phylum in all treatments.

The genera with relative abundance that was greater than 1% were considered dominant. For bacteria (Figure 4), excluding the unclassified genera, the CT treatment was dominated by *Woeseia* (3.69%). The SC treatment was dominated by *Woeseia* (5.46%) and *Sulfurovum* (2.10%). The MR treatment was dominated by *Woeseia* (4.11%), *Sulfurimonas* (3.99%), and *Sulfurovum* (2.68%).

The LEfSe analysis (Figure 5) showed that the relative abundance of some bacterial taxa differed significantly among the CT, SC, and MR treatments (*p* < 0.05, LDA > 3.0). At the phylum level, the relative abundance of Acidobacteria, Gemmatimonadetes, and Latescibacteria significantly increased in the SC treatment. The relative abundance of Bacteroidetes and Firmicutes significantly increased in the CT treatment. The relative abundance of Nitrospirae increased significantly in the MR treatment. At the genus level, excluding the unclassified genera, the relative abundance of *Robiginitalea* increased significantly in the SC treatment. The relative abundance of *Lutibacter*, *Maribacter*, *Maritimimonas*, *Defluviitaleaceae_UCG-011*, and *Fusibacter* increased significantly in the CT treatment. Excluding the unclassified genera, the relative abundance of *Sulfurovum* and *Lactobacillus* increased significantly in the MR treatment.

### 3.2. Fungal Diversity and Composition

There was no significant difference (*p* > 0.05) between Shannon and Chao1 diversity indices among all treatments (Figure 1). According to the Venn diagram (Figure 2), the three treatments shared 83 ASVs, while 490, 252, and 466 ASVs were unique to the CT, SC, and MR treatments, respectively. PCoA1 explained 18.63% of the variation and PCoA2 explained 12.15% of the variation (Figure 3). In total, 30.78% of variance was explained by the two principal coordinates. The results of PCoA showed that soil fungal community structure among the three treatments was significantly different (PERMANOVA test, *p* < 0.05), demonstrating that cutting plus tiling and mechanical rolling could shape the fungal community.

To explore differences in the fungal community structure under different *S. alterniflora* control methods, we compared the relative abundance of fungal communities for the three treatments. Ascomycota (16.09–32.07%), Chytridiomycota (2.55–25.4%), and Basidiomycota (4.33–10.77%) were the predominant phyla in all treatments (Figure 4). Ascomycota was the most predominant phylum in all treatments; its relative abundance was the highest in the MR treatment and lowest in the SC treatment. In addition, the relative abundance of Chytridiomycota was the highest in the MR treatment. However, Basidiomycota was less abundant in the MR treatment.

For fungi (Figure 4), excluding the unclassified genera, the CT treatment was dominated by *Bullera* (7.07%), *Knufia* (5.0%), and *Rhizophydium* (1.95%). The SC treatment was dominated by *Rhizophydium* (1.36%). The MR treatment was dominated by *Endocarpon* (2.25%) and *Alternaria* (1.68%).

The LEfSe analysis (Figure 6) showed the relative abundance of some fungal taxa differed among the CT, SC, and MR treatments. At the phylum level, excluding the unclassified phyla, no significant difference was observed. At the genus level, excluding the unclassified genera, the relative abundance of *Phaeococcomyces*, *Wickerhamomyces*, *Zygorhizidium*, and *Bullera* increased significantly in the CT treatment. The relative abundance of *Phoma*, *Myrothecium*, and *Nectria* increased significantly in the MR treatment.

## 4. Discussion

### 4.1. Effects of Different Invasion Control Methods on Soil Microbial Community Diversity

In this study, bacterial diversity significantly decreased in the CT treatment compared to the other two treatments, whereas the fungal diversity of the three treatments did not change significantly. Previous studies demonstrated that changes in quality and quantity of root exudates caused by changes in above-ground vegetation may result in a heterogeneous soil environment, which can further alter the soil microbial diversity [41,42,43]. In the coastal wetlands of eastern China, relative to native species, *S. alterniflora* invasion considerably triggers increases in root exudates and the soil C/N ratio because *S. alterniflora* has a higher photosynthetic efficiency and production [44]. Soil C/N ratio is an indicator of the availability of soil organic matter (SOM), and the decomposition rate of SOM decrease under the influence of higher C/N ratio [10,45,46]. Accumulated SOM and high root exudate concentration from *S. alterniflora* serve as a sources of soil microbial nutrition and energy and can thus promote the growth of soil microorganisms [46,47,48]. In addition, the developed and dense roots of *S. alterniflora* have a strong ability to promote siltation, which loosens the soil structure and increases soil water content [48]. Higher soil moisture is favorable for the formation of an anaerobic environment and promotes the growth of anaerobic microorganisms and the accumulation of organic matter [49].

The considerably decreased bacterial diversity and richness in the CT treatment can be explained by the nature of the control method; cutting removes the above-ground plants, which interrupts nutrient transfer from the plant to soil, while vigorous tilling also removes the roots situated in the ground, resulting in the reduction of root exudates [50]. Compared with the bacterial community, the fungal community appeared to be more tolerant to adverse soil conditions and may be able to utilize recalcitrant organic matter as a carbon source. Thus, fungal diversity and richness did not change substantially [2,51,52]. The decrease in soil bacterial diversity induced by cutting plus tilling likely affects soil ecosystem functioning, such as carbon and nitrogen cycling [53,54]. Mechanical rolling can increase soil bulk density and affect soil microbial community structure. Compared with low-density soil, the number of soil microorganisms decreases in compact soil, likely due to the latter’s poor soil aeration and the resulting inhibition of root growth, which limits root production [55,56]. Of note, in the present study, bacterial and fungal diversity and richness in the MR treatment did not differ significantly from the SC treatment. We speculate that the increased bulk density caused by mechanical rolling was not sufficient to limit soil microbial growth [57]. Within this range, the number of soil microorganisms is not closely associated with changes in the bulk density. Alternatively, mechanical rolling reduces the soil air capacity and thus promotes the growth of anaerobic bacteria [58].

### 4.2. Effects of Different Invasion Control Methods on Soil Microbial Community Structure

Soil bacterial and fungal community structures responded considerably to cutting plus tilling and mechanical rolling. The composition of dominant bacterial and fungal communities was not significantly different among the three treatments. However, the relative abundance of certain bacterial and fungal taxa differed significantly at different taxonomic levels among the treatments. In this study, Proteobacteria was the most predominant phylum among all bacterial phyla, which was generally consistent with the findings of previous studies [59,60]. The dominance and ubiquity of Proteobacteria in soil may be a result of their rapid growth rates and successful adaptation to environmental stresses [16,47,61]. Bacteroidetes and Firmicutes were more considerably abundant in the CT treatment. They exhibited particularly clear responses to cutting plus tilling, with severalfold higher relative abundance than those in the other two treatments. Bacteroidetes are copiotrophic microbes and they are closely related to r-strategy [62]. A previous study found that the increase of r-strategy bacteria would accelerate the decomposition of soil organic matter [43]. Firmicutes are demonstrated to be involved in the decomposition of organic matter [16]. Thus, we speculated that cutting plus tilling were not conductive to the accumulation of soil organic matter and further influenced soil carbon cycling. We found that the relative abundance of Nitrospirae increased significantly in the MR treatment. The phylum Nitrospirae is a nitrifying bacterial group which can participate in nitrogen cycling [63]. Therefore, mechanical rolling may have a positive effect on the nitrogen cycling. For fungi, the dominant phyla were Ascomycota, Chytridiomycota, and Basidiomycota. No significant differences was observed in their relative abundance among treatments. Basidiomycota and Ascomycota are often the dominant members of soil fungal communities [64,65]. Ascomycota are more capable of resisting environmental stress and utilizing multiple resources, which may be the reason why they are predominant in soil [66]. Ascomycota are considered one of the most important decomposers in soil and are oligotrophs, which enables their survival in environments with low resource availability [67]. By contrast, Basidiomycota prefer environments with high fertility [67]. Furthermore, Basidiomycota play a crucial role in decomposing recalcitrant materials (e.g., lignin and cellulose) [43].

At the genus level, the relative abundance of *Desulfobulbus*, *Thioalkalispira*, and *Sulfurovum* differed significantly in the CT, SC, and MR treatments (Appendix A). The relative abundance of *Sulfurovum* and *Desulfobulbus* increased significantly in the MR treatment. Additionally, the relative abundance of *Thioalkalispira* was the highest in the SC treatment. *Desulfobulbus* are sulfate-reducing bacteria, and *Sulfurovum* and *Thioalkalispira* are sulfur-oxidizing bacteria, which can respectively drive the reduction and oxidation reactions in the sulfur cycling [68]. *Desulfobulbus* also have a considerable impact on carbon cycling [63]. The sulfate-reducing bacteria tend to live in anoxic environments [69]. Rolling can inhibit soil aeration, which may explain the significant increase of *Desulfobulbus* in the MR treatment.

## 5. Conclusions

Different invasion control methods caused different shifts in soil bacterial and fungal community structures and diversity. After cutting plus tilling, the diversity and richness of the bacterial community significantly decreased; however, no significant changes were noted in fungal diversity and richness. Such shifts in soil bacterial and fungal community structures and diversity would affect soil ecosystem stability. In particular, the relative abundance of Bacteroidetes and Firmicutes significantly increased, which may cause the rapid decomposition of soil organic matter. The relative abundance of genera involved in sulfur cycling significantly changed, which likely alters soil nutrient cycling rates. After mechanical rolling, there were no significant differences in microbial diversity, but bacterial and fungal community structures considerably differed. The relative abundance of bacterial taxa associated with nutrient cycling increased considerably, which may promote soil ecological processes. These results provide useful insights into responses of bacterial and fungal communities and the potential ecological consequences of *S. alterniflora* invasion control methods. In the future, based on this study, we will investigate the microbial information of the non-invaded site, and combine the relationship between soil physicochemical properties and soil microbial communities to provide a theoretical basis for the protection of soil ecosystems.

## Figures and Tables

**Figure 1 microorganisms-10-01122-f001:**
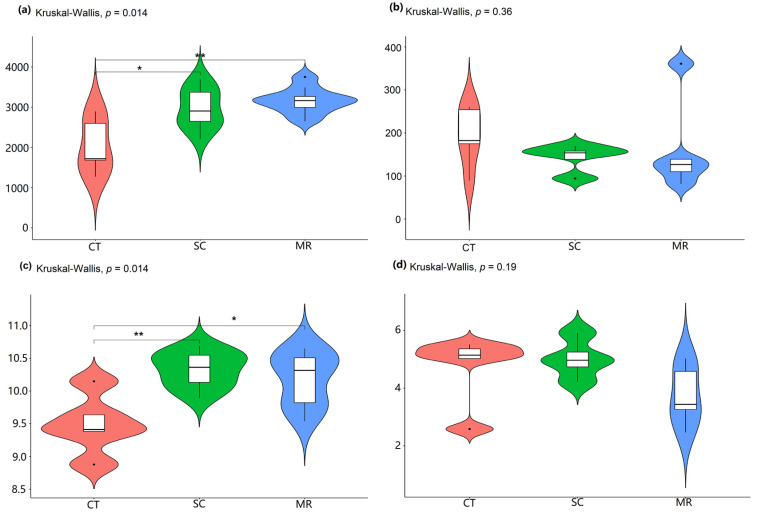
Differences in alpha diversity indices of bacteria ((**a**). Chao1; (**c**). Shannon) and fungi ((**b**). Chao1; (**d**). Shannon). One asterisk (*) represents significant difference (*p* < 0.05) and two asterisks (**) represent highly significant difference (*p* < 0.01) between treatments.

**Figure 2 microorganisms-10-01122-f002:**
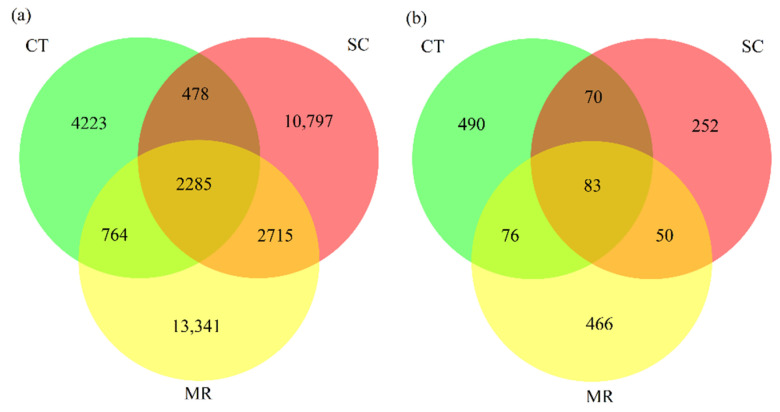
Venn diagrams depicting the number of shared and unique (**a**) bacterial and (**b**) fungal amplicon sequence variants (ASVs) among treatments. Each circle represents sampled compartments. Values within intersections represent shared ASVs, values outside intersections represent unique ASVs.

**Figure 3 microorganisms-10-01122-f003:**
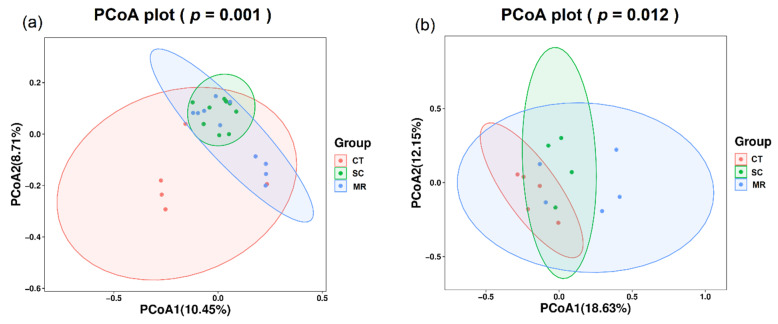
Principal coordinates analysis (PCoA) with unweighted UniFrac distance of (**a**) bacterial and (**b**) fungal community composition. Different colored ellipsoids represent different treatments.

**Figure 4 microorganisms-10-01122-f004:**
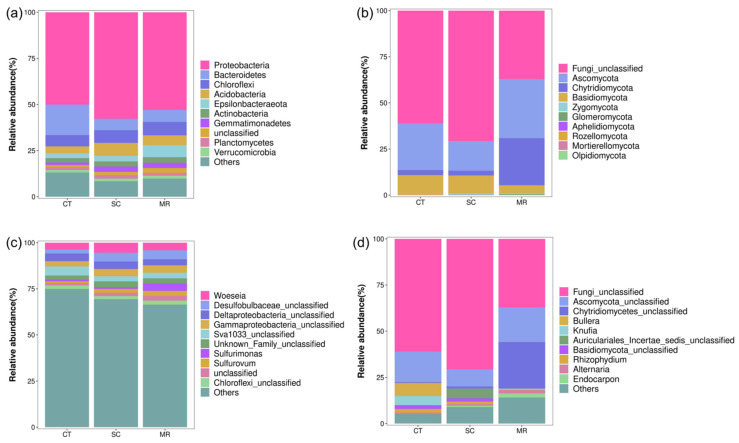
Bacterial (**a**,**c**) and fungal (**b**,**d**) community composition at the phylum (**a**,**b**) and genus (**c**,**d**) level.

**Figure 5 microorganisms-10-01122-f005:**
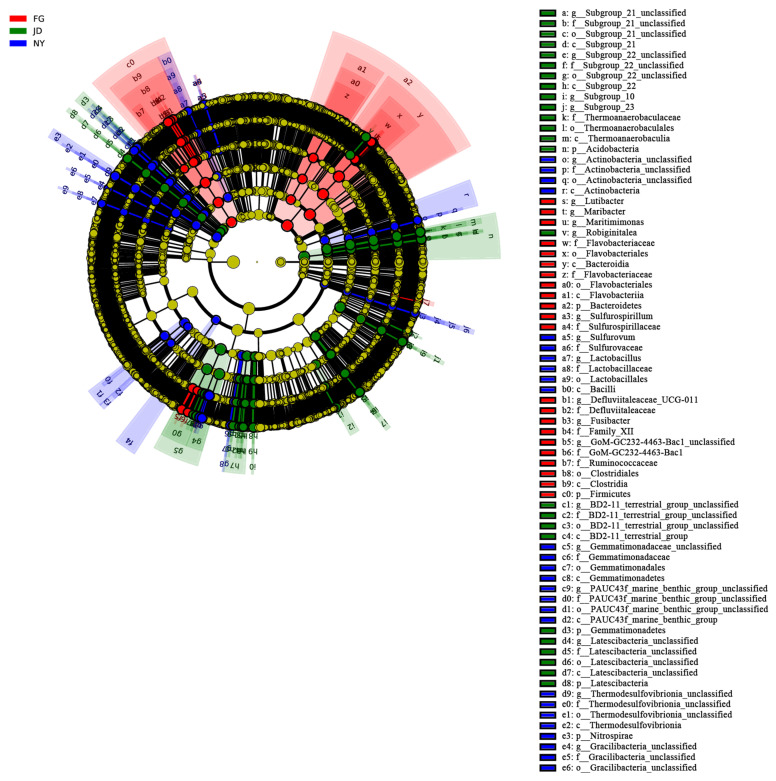
LEfSe showing the significant differences at different bacterial taxonomic levels among the CT, SC, and MR treatments. Different colored dots represent the taxa with significant differences among all treatments. The inner to outer circles represent taxa from phylum to species.

**Figure 6 microorganisms-10-01122-f006:**
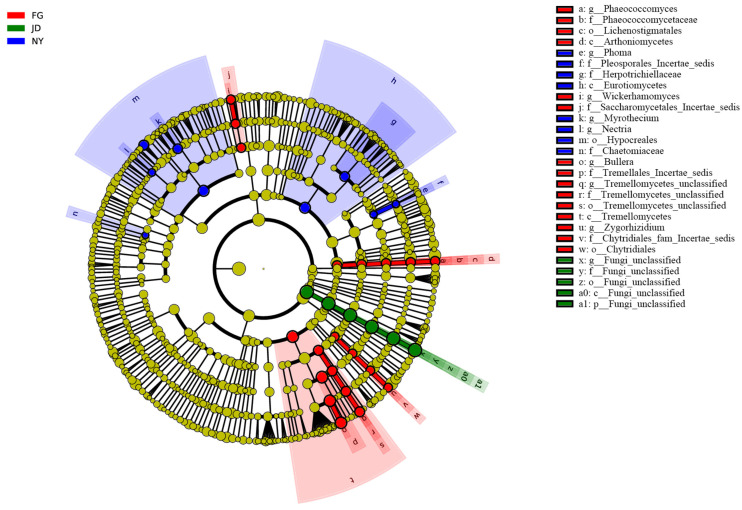
LEfSe showing the significant differences at different fungal taxonomic levels among the CT, SC, and MR treatments. Different colored dots represent the taxa with significant differences among all treatments. The inner to outer circles represent taxa from phylum to species.

## Data Availability

The data that support the findings of this study are available in NCBI (https://www.ncbi.nlm.nih.gov/ (accession number (PRJNA789127 and PRJNA789435))).

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
