# Peer review of "Responses of Soil Microbiota to Different Control Methods of the Spartina alterniflora in the Yellow River Delta"

_microorganisms, 2022, doi:10.3390/microorganisms10061122_

Round 1

Reviewer 1 Report

The manuscript by Li et al., studied the effect of two different invasion control methods of Spartina alterniflora on soil microbial communities, in particular bacteria and fungi. Using high throughput sequencing technology, the authors showed that bacterial diversity and richness in the cutting plus tilling group reduced substantially but fungal diversity and richness did not show any remarkable change. On the other hand, bacterial and fungal diversity and richness in the mechanical rolling treatment were not affected considerably. Based on their results, the authors finally concluded that mechanical rolling is a relatively suitable method for controlling S. alterniflora invasion. Although these results will contribute to better understand two different invasion control methods, several improvements must be done before the acceptance of this manuscript.

Major comments:

Although I agree that the knowledge regarding soil microbial communities it is important and could give insights about soil health, I do not think that by itself the information present in this manuscript is enough to conclude which of these two methods is better to control S. alterniflora invasion. Several parameters such as secondary invasion and price of the different methods will have a more pivotal role at the moment of selection. For example, if one method does not affect microbial communities but it is not effective at preventing secondary invasion, this method would not be effective. On the other hand, how these control methods affect the colonization of endemic plants? Moreover, this manuscript would have been more informative if information regarding the physicochemical and geochemical conditions of the environment would have been presented and correlated with the information of microbial communities. Since no information of total carbon, nitrogen or other important parameters were measured, the information regarding the potential effects of the different genus detected is just speculative and do not necessarily reflect what is occurring on the environment.

The authors provided a good background regarding S. alterniflora invasion. It was mentioned that S. alterniflora can change the physicochemical conditions of the soil and therefore the microbial communities present. Then the authors compared the microbial communities present after two different invasion control methods and a non-treated soil. Since S. alterniflora change the microbial communities present, why the results were only compared with a non-treated soil invaded with S. alterniflora and not also with a soil not invaded by this plant? This is important to really know the effect of the different invasion control methods on the native microbial communities. The way it is presented now will allow us to compare the microbial communities present in a soil invaded with S. alterniflora but not with those microbial communities that should be really present in this environment. Why should we care to preserve the microbial communities that are present in soils invaded with S. alterniflora?  

Regarding methodology, why these methods were compared and no other invasion control methods? Were the DNA extractions made from triplicate samples and then pooled before PCR? If not, please explain why. Were the PCR made in triplicate and then pooled before sequenciation? If not, please explain why. Why archaea was not studied? How long after the treatments were the samples collected. Please add more information

Minor comments

Line 14: I would delete “(3)Methods:”

Line 30: Genus and specie should be italicized. Please fix this throughout the manuscript

Linre 35: change “;” by “.”

Line 73: “et al” should be italicized

Lines 97-99: Although knowing how these controls methods affect microbial communities and in consequence the functioning of these ecosystems is interesting, the final decision regarding the control method to be used will depend more in the efficacy of the control and if this method is enough to avoid secondary invasion.

Lines 106-107: Are there previous studies about bacterial or fungal communities in that area?

Lines 111-112: Why these abbreviations were used? To me, it makes no sense (and difficulted the reading of the manuscript) to call FG, NY and JD the cutting plus tilling, mechanical rolling and non-controlled treatment respectively. I would suggest to rename these abbreviations using C+T, MR and NC.

Lines 112-1118: Why the cutting plus tilling treatment was performed 3 times and the mechanical rolling only once?

Line 118: why only 5 plots were used in the FG treatment?

Line 120-122: how long after the treatment were the samples collected. How much sample was collected? Please add more information

Line 123: brought back where? Please add more information

Line 139: Add space between 50 and ng

Figure 1: Please fix the legend inside the figure because it is hard to read. The authors should increase font size and color of plot numbers to make them readable

Figure 2: Please increase font size

Figure 3: Please increase font size

Figure 5: Please increase font size. As it is now it is useless

Lines 215-232: Genus should be italicized

Lines 218-221: Subgroups and groups are not valid genus. Please explain

Line 289: Add space between “content” and [48]

Lines 291-293: It is unclear why the authors concluded this, considering that the bacterial diversity did not differ significantly between the NY and the JD treatments (lines 173-174)

Line 292: please add “is” before “attributed”

All genus and species names should be italicized

Line 372: Which supporting information?

Author Response

    We appreciate editor and reviewers very much for your valuable and constructive comments on our manuscript entitled “Responses of soil microbiota to different control methods of the Spartina alterniflora in the Yellow River delta”. We have studied the comments of the reviewers carefully and tried our best to revise the manuscript according to the Reviewers’ good comments. Revised portion are marked in red in the manuscript. We also enlisted the help of native English editors. We appreciate for Editors and Reviewers’ warm work earnestly, and hope that the revision will meet with approval. Look forward to hearing from you.

Yours sincerely

Shuai Shang

To Reviewer 1

Comments and Suggestions for Authors

The manuscript by Li et al., studied the effect of two different invasion control methods of Spartina alterniflora on soil microbial communities, in particular bacteria and fungi. Using high throughput sequencing technology, the authors showed that bacterial diversity and richness in the cutting plus tilling group reduced substantially but fungal diversity and richness did not show any remarkable change. On the other hand, bacterial and fungal diversity and richness in the mechanical rolling treatment were not affected considerably. Based on their results, the authors finally concluded that mechanical rolling is a relatively suitable method for controlling S. alterniflora invasion. Although these results will contribute to better understand two different invasion control methods, several improvements must be done before the acceptance of this manuscript.

Major comments:

Although I agree that the knowledge regarding soil microbial communities it is important and could give insights about soil health, I do not think that by itself the information present in this manuscript is enough to conclude which of these two methods is better to control S. alterniflora invasion. Several parameters such as secondary invasion and price of the different methods will have a more pivotal role at the moment of selection. For example, if one method does not affect microbial communities but it is not effective at preventing secondary invasion, this method would not be effective. On the other hand, how these control methods affect the colonization of endemic plants? Moreover, this manuscript would have been more informative if information regarding the physicochemical and geochemical conditions of the environment would have been presented and correlated with the information of microbial communities. Since no information of total carbon, nitrogen or other important parameters were measured, the information regarding the potential effects of the different genus detected is just speculative and do not necessarily reflect what is occurring on the environment.

Response: Thanks for the suggestions. Conclusions regarding which of the two methods is better to control have been deleted throughout the manuscript. Actually, we measured basic soil parameters (such as pH, salinity, temperature) in this experiment. However, we found that due to the short distance between sampling sites and limited investigation area, the variations of mentioned physichemical properties were not obvious. Besides, the habitats as well as vegetation conditions tended to be almost the same. Thus, regrettably, we feel sorry that we missed measuring some other physicochemical parameters of the soil that we considered as minor factors after sampling. Meanwhile, our study aimed to explore the influence of control methods on soil microbial communities, focusing on the changes in soil microbial community diversity and structure. This study is expected to provide basic microbiological information for the investigation of the S. alterniflora invasion control methods. In the future, we will integrate more soil physicochemical data with soil microbial communities to carry out more profound researches.

The authors provided a good background regarding S. alterniflora invasion. It was mentioned that S. alterniflora can change the physicochemical conditions of the soil and therefore the microbial communities present. Then the authors compared the microbial communities present after two different invasion control methods and a non-treated soil. Since S. alterniflora change the microbial communities present, why the results were only compared with a non-treated soil invaded with S. alterniflora and not also with a soil not invaded by this plant? This is important to really know the effect of the different invasion control methods on the native microbial communities. The way it is presented now will allow us to compare the microbial communities present in a soil invaded with S. alterniflora but not with those microbial communities that should be really present in this environment. Why should we care to preserve the microbial communities that are present in soils invaded with S. alterniflora?  

Response: Thanks for the suggestion. In this study, we focused on how soil bacterial community changed after the implement of control methods. So, to some distance, we aimed to evaluate the control effects of different treating measures. In addition, this study is part of our project, in which we also conducted similar studies in this area (Shang et al.), and we will monitor this area for a long time. In the future, undoubtedly, we will also investigate the microbial communities of soil not invaded with S. alterniflroa. This is very meaningful.

Shang, S., et al. "Effects of Spartina Alterniflora Invasion on the Community Structure and Diversity of Wetland Soil Bacteria in the Yellow River Delta." Ecol Evol 12.5 (2022): e8905. Print.

Regarding methodology, why these methods were compared and no other invasion control methods? Were the DNA extractions made from triplicate samples and then pooled before PCR? If not, please explain why. Were the PCR made in triplicate and then pooled before sequenciation? If not, please explain why. Why archaea was not studied? How long after the treatments were the samples collected. Please add more information

Response: Thanks for the suggestion. In our previous field exploration, we found that a lower rate of S. alterniflora regrowth following deliberate or unintentional artificial rolling and cutting plus tilling. Having identified this phenomenon, we made the hypothesis and therefore designed the present experiment for the two control methods. The control and management of S. alterniflora is a long-term and difficult task, and in subsequent studies we will also explore the effects of other control methods on the soil microbial communities. The DNA extractions and PCR products were from triplicate samples and they were pooled before the next procedure, respectively. This information has been added in line 140-141. We read literatures and found that there are studies selecting bacteria and fungi as representatives of microbial communities (Cao et al.; Zhang, Bai, Jia, et al.; Jiang et al.), so we chose bacteria and fungi in this study. Archaea are also the main members of the soil microbial community, and we will consider studying them in the future research. The measures of cutting plus tilling and mechanical rolling took a long time to implement. After the condition of the sampling sites was stabilized (about one week), we collected samples. This information has been added in line 119-120.

Cao, M., et al. "Effects of Spartina Alterniflora Invasion on Soil Microbial Community Structure and Ecological Functions." Microorganisms 9.1 (2021). Print.

Jiang, Shuai, et al. "Changes in Soil Bacterial and Fungal Community Composition and Functional Groups During the Succession of Boreal Forests." Soil Biology and Biochemistry 161.3 (2021): 108393. Print.

Zhang, G., et al. "Shifts of Soil Microbial Community Composition Along a Short-Term Invasion Chronosequence of Spartina Alterniflora in a Chinese Estuary." Science of the Total Environment 657 (2019): 222-33. Print.

Minor comments

Line 14: I would delete “(3)Methods:”

Response: Thanks for the suggestion. “(3)Methods:” has been deleted.

Line 30: Genus and specie should be italicized. Please fix this throughout the manuscript

Response: Thanks for the suggestion. Genus and species have been italicized throughout the manuscript.

Linre 35: change “;” by “.”

Response: Thanks for the suggestion. “;” has been changed to “.” 

Line 73: “et al” should be italicized

Response: Thanks for the suggestion. “et al” has been italicized.

Lines 97-99: Although knowing how these controls methods affect microbial communities and in consequence the functioning of these ecosystems is interesting, the final decision regarding the control method to be used will depend more in the efficacy of the control and if this method is enough to avoid secondary invasion.

Response: Thanks for the suggestion. The original sentences “We also assessed......functioning of wetland ecosystems” have been changed to “The results of this study may provide a microbiological basis for evaluating the impact of control measures on soil ecosystems”.

Lines 106-107: Are there previous studies about bacterial or fungal communities in that area?

Response: There are previous studies about bacterial or fungal communities in the Yellow River Delta (Li et al.; Zhang, Bai, Tebbe, et al.).

Li, J., et al. "Influence of Plants and Environmental Variables on the Diversity of Soil Microbial Communities in the Yellow River Delta Wetland, China." Chemosphere 274 (2021): 129967. Print.

Zhang, G., et al. "Spartina Alterniflora Invasions Reduce Soil Fungal Diversity and Simplify Co-Occurrence Networks in a Salt Marsh Ecosystem." Science of the Total Environment 758 (2021): 143667. Print.

Zhang, G., et al. "Shifts of Soil Microbial Community Composition Along a Short-Term Invasion Chronosequence of Spartina Alterniflora in a Chinese Estuary." Science of the Total Environment 657 (2019): 222-33. Print.

Lines 111-112: Why these abbreviations were used? To me, it makes no sense (and difficulted the reading of the manuscript) to call FG, NY and JD the cutting plus tilling, mechanical rolling and non-controlled treatment respectively. I would suggest to rename these abbreviations using C+T, MR and NC.

Response: Thanks for the suggestion. “non-controlled treatment” has been changed to “statistical control”, and the abbreviations have changed to “CT”, “SC” and “MR” to call the cutting plus tilling, statistical control and mechanical rolling respectively.

Lines 112-1118: Why the cutting plus tilling treatment was performed 3 times and the mechanical rolling only once?

Response: Thanks for the suggestion. The two measures were implemented the same number of times, and this information was mentioned in line 113-115.

Line 118: why only 5 plots were used in the FG treatment?

Response: Thanks for the suggestion. I was sorry that I made the mistake. “5” has been changed to “10”. We collected ten samples from each treatment group respectively.

Line 120-122: how long after the treatment were the samples collected. How much sample was collected? Please add more information

Response: Thanks for the suggestion. “Samples were collected one week after the end of the treatment (the condition of the treated sites stabilized).” has been added in line 119-120. A total of 30 samples were collected. The information has been added in line 124.

Line 123: brought back where? Please add more information

Response: Thanks for the suggestion. “to the laboratory” has been added after “brought back” in line 125.

Line 139: Add space between 50 and ng

Response: Thanks for the suggestion. Space has been added between “50” and “ng”.

Figure 1: Please fix the legend inside the figure because it is hard to read. The authors should increase font size and color of plot numbers to make them readable

Figure 2: Please increase font size

Figure 3: Please increase font size

Figure 5: Please increase font size. As it is now it is useless

Response: Thanks for the suggestion. Figure 1 has been fixed and put in the supplementary materials. The other figures (Fig.2, Fig.3 and Fig.5)have also been fixed.

Lines 215-232: Genus should be italicized

Response: Thanks for the suggestion. Genus has been italicized throughout the manuscript.

Lines 218-221: Subgroups and groups are not valid genus. Please explain

Response: Thanks for the suggestion. Subgroups and groups have been deleted throughout the manuscript.

Line 289: Add space between “content” and [48]

Response: Thanks for the suggestion. Space has been added between “content” and [48] in line 286.

Lines 291-293: It is unclear why the authors concluded this, considering that the bacterial diversity did not differ significantly between the NY and the JD treatments (lines 173-174)

Response: Thanks for the suggestion. The sentence has been deleted in line 288.

Line 292: please add “is” before “attributed”

Response: Thanks for the suggestion. The sentence has been deleted.

All genus and species names should be italicized

Response: Thanks for the suggestion. All genus and species name have been italicized.

Line 372: Which supporting information?

Response: Thanks for the suggestion, in the revised manuscript, we have add the supporting information site.

Reviewer 2 Report

The reviewed manuscript has practical relevance, yielding interesting applications, worth publishing, once the authors address some aspects, which in my opinion require their attention.

The article is overall well-argued and written. Despite this, there are some minor syntactic issues, I mention some indicative ones below:

  • Page 1, line 28, “Coastal wetlands, the transition zones…” should read “Coastal wetlands, transition zones…”
  • Page 3, lines 109-110, “We set up three treatment blocks…” should read “Three treatment blocks were set up…”

There are a few more of these issues throughout the text and I’d ask the authors to carefully proofread their manuscript again or consult a colleague versed in scientific English editing.

There are also some instances of run-on sentences and random comma placement, often confusing the primary points of the sentence.

Some additional comments follow:

  • Figure 1 would be better suited in as a supplementary figure
  • Please elaborate on what the “non-controlled treatment” encompasses.
  • A graphical synopsis of the methodology employed would assist as a quick overview
  • As some readers might not be familiar with some procedures (even commercial available ones like the Z.N.A.®Soil 128 DNA Kit) these should be briefly explained
  • The text within figure 5 is unreadable, due to size and quality of the image.
  • The phrase of line 255 is a repetition of line 215 and thus redundant
  • The statement “In this study, compared to the JD treatment, the diversity and richness of soil bacte- rial and fungal communities differed in the FG and NY treatments.” (lines 275-276) does not seem to be in line with the one of lines 173-174.

Author Response

    We appreciate editor and reviewers very much for your valuable and constructive comments on our manuscript entitled “Responses of soil microbiota to different control methods of the Spartina alterniflora in the Yellow River delta”. We have studied the comments of the reviewers carefully and tried our best to revise the manuscript according to the Reviewers’ good comments. Revised portion are marked in red in the manuscript. We also enlisted the help of native English editors. We appreciate for Editors and Reviewers’ warm work earnestly, and hope that the revision will meet with approval. Look forward to hearing from you.

Yours sincerely

Shuai Shang

To Reviewer 2

Comments and Suggestions for Authors

The reviewed manuscript has practical relevance, yielding interesting applications, worth publishing, once the authors address some aspects, which in my opinion require their attention.

The article is overall well-argued and written. Despite this, there are some minor syntactic issues, I mention some indicative ones below:

Page 1, line 28, “Coastal wetlands, the transition zones…” should read “Coastal wetlands, transition zones…”

Response: Thanks for the suggestion. “Coastal wetlands, the transition zones…” has been changed to “Coastal wetlands, transition zones…”

Page 3, lines 109-110, “We set up three treatment blocks…” should read “Three treatment blocks were set up…”

Response: Thanks for the suggestion. “We set up three treatment blocks…” has been changed to “Three treatment blocks were set up…” in line110.

There are a few more of these issues throughout the text and I’d ask the authors to carefully proofread their manuscript again or consult a colleague versed in scientific English editing. There are also some instances of run-on sentences and random comma placement, often confusing the primary points of the sentence.

Response: Thanks for the suggestion. The similar issues have been amended in the manuscript.

Some additional comments follow:

Figure 1 would be better suited in as a supplementary figure

Response: Thanks for the suggestion. Figure 1 has been put in the supplementary materials.

Please elaborate on what the “non-controlled treatment” encompasses.

Response: Thanks for the suggestion. “non-controlled treatment” has been changed to “statistical control without any real treatment” in line 112-113.

A graphical synopsis of the methodology employed would assist as a quick overview. As some readers might not be familiar with some procedures (even commercial available ones like the Z.N.A.®Soil 128 DNA Kit) these should be briefly explained

Response: Thanks for the suggestions. A graphical synopsis of the methodology employed would assist as a quick overview. While, we read literatures and found that most studies tended to describe the whole procedure in words and did not explain much about the kit and so on (Cao et al.; Jiang et al.; Qu et al.).

Cao, M., et al. "Effects of Spartina Alterniflora Invasion on Soil Microbial Community Structure and Ecological Functions." Microorganisms 9.1 (2021). Print.

Jiang, Shuai, et al. "Changes in Soil Bacterial and Fungal Community Composition and Functional Groups During the Succession of Boreal Forests." Soil Biology and Biochemistry 161.3 (2021): 108393. Print.

Qu, Z., et al. "The Response of the Soil Bacterial Community and Function to Forest Succession Caused by Forest Disease." Functional Ecology 34.12 (2020): 2548-59. Print.

The text within figure 5 is unreadable, due to size and quality of the image.

Response: Thanks for the suggestion. The figure 5 (figure 4 now) has been fixed.

The phrase of line 255 is a repetition of line 215 and thus redundant

Response: Thanks for the suggestion. The sentence in line 255 has been deleted.

The statement “In this study, compared to the JD treatment, the diversity and richness of soil bacte- rial and fungal communities differed in the FG and NY treatments.” (lines 275-276) does not seem to be in line with the one of lines 173-174.

Response: Thanks for the suggestion. This sentence has been changed to “bacterial diversity significantly decreased in the CT treatment compared to the other two treatments, whereas the fungal diversity of the three treatments did not change significantly.” in line 272-274.

Reviewer 3 Report

General: S. alterniflora and Spartina alterniflora should be in italics wherever they occur (lines 30, 31–32, 36 and many other places). In addition, there are other species names mentioned throughout the manuscript. They need to be put into italics.

   A major problem with the current manuscript is that the study lacks replication. That is, there is only a single 'replicate' of each of the three 'treatments', which includes the statistical control JD, which was untreated. Each of the 3 treatment 'blocks' in the Guangli River was subdivided into 5 or 10 plots, each 1 sq. m. and separated by 100 m. In statistical parlance, this constitutes 'pseudoreplication'. Although some scientists totally disparage pseudoreplication, I am not one to condemn it, as I recognize that it is virtually impossible to carry out an experiment involving large masses of material in any other way. True biological replication would involve an impractically large experimental program. However, pseudoreplication does limit the generality of the conclusions. It is not correct to imply in the Abstract that "Overall, mechanical rolling is a relatively suitable method for controlling S. alterniflora invasion". This study provided evidence for this statement only from a single treatment block; many more studies may be needed before such a conclusion can be justified.

Specific:

Line 14. Delete “(3) Methods:”

Line 44. Insert a comma after “wetlands”

Line 107. The species should read “Phragmites australis (Cav.) Tran. ex Stead.” That is, it is australis, not australias, and the full stops after Cav, Trin and Steud are needed, as these are abbreviations of longer names. Similarly, it should be “Lour.”, not “Lour”

Line 112. Replace "non-controlled treatment" by "statistical control". Confusion arises because you are using 'control' to refer to biological control of the Spartina invasion. In statistics, control is used to mean a  level of a treatment level in which no real treatment or procedure was applied to the material under study. It is in that sense that 'statistical control' would be understood by the readers.

Line 116. It should read “machine boats”, not “the machine boats”. By the way, what are machine boats? I couldn’t find any reference to that term when I did an Internet search.

Line 129. It should read “manufacturer’s”, not “manufactures’s”

Line 133. Insert “the” between “of” and “fungal”

Line 143. It should read “paired-end’, not “paried-end”

Line 183. Replace “extremely” by “highly”. P< 0.01 is not extreme, and the expression  “highly significant” is commonly used in the statistical literature.

Line 196. It should read “structure … was”, not “structure … were”

Line 244. It should read “structure … was”, not “structure … were”

Line 245, et seq. It should be made clear in this paragraph that the JD “treatment” is really a statistical control, and that the soil did not undergo any real treatment.

Line 295. “Cutting” should read “cutting”

Line 333. It should read “Nitrospirae is”, not “Nitrospirae are”.

Line 338. “are” is preferable to “were”

Line 359. It should be “the bacterial community”

Lines 386–388. Supply Acknowledgments or delete this section.

Author Response

    We appreciate editor and reviewers very much for your valuable and constructive comments on our manuscript entitled “Responses of soil microbiota to different control methods of the Spartina alterniflora in the Yellow River delta”. We have studied the comments of the reviewers carefully and tried our best to revise the manuscript according to the Reviewers’ good comments. Revised portion are marked in red in the manuscript. We also enlisted the help of native English editors. We appreciate for Editors and Reviewers’ warm work earnestly, and hope that the revision will meet with approval. Look forward to hearing from you.

Yours sincerely

Shuai Shang

To Reviewer 3

Comments and Suggestions for Authors

General: S. alterniflora and Spartina alterniflora should be in italics wherever they occur (lines 30, 31–32, 36 and many other places). In addition, there are other species names mentioned throughout the manuscript. They need to be put into italics.

Response: Thanks for the suggestion. “S. alterniflora”, “Spartina alterniflora” and other species name have been italicized throughout the manuscript.

 A major problem with the current manuscript is that the study lacks replication. That is, there is only a single 'replicate' of each of the three 'treatments', which includes the statistical control JD, which was untreated. Each of the 3 treatment 'blocks' in the Guangli River was subdivided into 5 or 10 plots, each 1 sq. m. and separated by 100 m. In statistical parlance, this constitutes 'pseudoreplication'. Although some scientists totally disparage pseudoreplication, I am not one to condemn it, as I recognize that it is virtually impossible to carry out an experiment involving large masses of material in any other way. True biological replication would involve an impractically large experimental program. However, pseudoreplication does limit the generality of the conclusions. It is not correct to imply in the Abstract that "Overall, mechanical rolling is a relatively suitable method for controlling S. alterniflora invasion". This study provided evidence for this statement only from a single treatment block; many more studies may be needed before such a conclusion can be justified.

Response: Thanks for the suggestion. In the present study, we conducted an exploratory experiment and selected a representative area to carry out our investigations. In future studies, we will carry out more in-depth studies from a larger spatial scale. The conclusions regarding which control methods is better to control throughout the manuscript has been deleted. This study focused on the influence of different control methods on soil microbial communities.

Specific:

Line 14. Delete “(3) Methods:”

Response: Thanks for the suggestion. “(3) Methods:” has been deleted.

Line 44. Insert a comma after “wetlands”

Response: Thanks for the suggestion. The comma has been inserted after “wetlands”.

Line 107. The species should read “Phragmites australis (Cav.) Tran. ex Stead.” That is, it is australis, not australias, and the full stops after Cav, Trin and Steud are needed, as these are abbreviations of longer names. Similarly, it should be “Lour.”, not “Lour”

Response: Thanks for the suggestion. “Phragmites australias Trin” Has been changed to “Phragmites australis (Cav.) Tran. ex Stead.” “Lour” has been changed to “Lour.”

Line 112. Replace "non-controlled treatment" by "statistical control". Confusion arises because you are using 'control' to refer to biological control of the Spartina invasion. In statistics, control is used to mean a  level of a treatment level in which no real treatment or procedure was applied to the material under study. It is in that sense that 'statistical control' would be understood by the readers.

Response: Thanks for the suggestion. “non-controlled treatment” has been replaced by “statistical control”.

Line 116. It should read “machine boats”, not “the machine boats”. By the way, what are machine boats? I couldn’t find any reference to that term when I did an Internet search.

Response: Thanks for the suggestion. I was sorry that I used the wrong word. “the machine boats” has been changed to “ploughing boat”.

Line 129. It should read “manufacturer’s”, not “manufactures’s”

Response: Thanks for the suggestion. “manufactures’s” has been changed to “manufacturer’s”.

Line 133. Insert “the” between “of” and “fungal”

Response: Thanks for the suggestion. “the” has been inserted between “of” and “fungal”.

Line 143. It should read “paired-end’, not “paried-end”

Response: Thanks for the suggestion. “paried-end” has been changed to “paired-end” in line 145.

Line 183. Replace “extremely” by “highly”. P< 0.01 is not extreme, and the expression  “highly significant” is commonly used in the statistical literature.

Response: Thanks for the suggestion. “extremely” has been change to “highly” in line  185.

Line 196. It should read “structure … was”, not “structure … were”

Response: Thanks for the suggestion. “structure … were” has been changed to “structure … was” in line 199.

Line 244. It should read “structure … was”, not “structure … were”

Response: Thanks for the suggestion. “structure … were” has been changed to “structure … was” in line 244.

Line 245, et seq. It should be made clear in this paragraph that the JD “treatment” is really a statistical control, and that the soil did not undergo any real treatment.

Response: Thanks for the suggestion. JD treatment has been changed to “statistical control without any real treatment” in line 112-113.

Line 295. “Cutting” should read “cutting”

Response: Thanks for the suggestion. “Cutting” has been change to “cutting” in line 291.

Line 333. It should read “Nitrospirae is”, not “Nitrospirae are”.

Response: Thanks for the suggestion. “Nitrospirae are” has been changed to “Nitrospirae is” in line 326.

Line 338. “are” is preferable to “were”

Response: Thanks for the suggestion. “are” has been changed to “were” line 331.

Line 359. It should be “the bacterial community”

Response: Thanks for the suggestion. “bacterial community” has been change to “the bacterial community” in line 35.

Lines 386–388. Supply Acknowledgments or delete this section.

Response: Thanks for the suggestion. We have delete this section in the revised manuscript.

Round 2

Reviewer 1 Report

The authors did a good job improving their manuscript. In this revised version of their manuscript it is more evident that these results allow to have an idea  of the effect of these different control methods on the microbial communities (but do not imply which method is better)